# Immune Responses to Orally Administered Recombinant *Lactococcus lactis* Expressing Multi-Epitope Proteins Targeting M Cells of Foot-and-Mouth Disease Virus

**DOI:** 10.3390/v13102036

**Published:** 2021-10-09

**Authors:** Fudong Zhang, Zhongwang Zhang, Xian Li, Jiahao Li, Jianliang Lv, Zhongyuan Ma, Li Pan

**Affiliations:** 1State Key Laboratory of Veterinary Etiological Biology, National Foot-and-Mouth Diseases Reference Laboratory, Lanzhou Veterinary Research Institute, Chinese Academy of Agricultural Sciences, Lanzhou 730046, China; zhangfudongg@163.com (F.Z.); zhangzhongwang@caas.cn (Z.Z.); lixian0209@163.com (X.L.); lijiahao970421@163.com (J.L.); lvjianliang@caas.cn (J.L.); 18109445816@163.com (Z.M.); 2Jiangsu Co-Innovation Center for Prevention and Control of Important Animal Infectious Diseases and Zoonoses, Yangzhou 225009, China

**Keywords:** FMDV, *L. lactis*, mucosal immunity, multi-epitope, targeting ligand

## Abstract

Foot and mouth disease virus (FMDV), whose transmission occurs through mucosal surfaces, can also be transmitted through aerosols, direct contact, and pollutants. Therefore, mucosal immunity can efficiently inhibit viral colonization. Since vaccine material delivery into immune sites is important for efficient oral mucosal vaccination, the M cell-targeting approach is important for effective vaccination given M cells are vital for luminal antigen influx into the mucosal lymph tissues. In this study, we coupled M cell-targeting ligand Co1 to multi-epitope TB1 of FMDV to obtain TB1-Co1 in order to improve delivery efficiency of the multi-epitope protein antigen TB1. *Lactococcus lactis* (*L. lactis*) was engineered to express heterologous antigens for applications as vaccine vehicles with the ability to elicit mucosal as well as systemic immune responses. We successfully constructed *L. lactis* (recombinant) with the ability to express multi-epitope antigen proteins (TB1 and TB1-Co1) of the FMDV serotype A (named *L. lactis*-TB1 and *L. lactis*-TB1-Co1). Then, we investigated the immunogenic potential of the constructed recombinant *L. lactis* in mice and guinea pigs. Orally administered *L. lactis*-TB1 as well as *L. lactis*-TB1-Co1 in mice effectively induced mucosal secretory IgA (SIgA) and IgG secretion, development of a strong cell-mediated immune reactions, substantial T lymphocyte proliferation in the spleen, and upregulated IL-2, IFN-γ, IL-10, and IL-5 levels. Orally administered ligand-conjugated TB1 promoted specific IgG as well as SIgA responses in systemic and mucosal surfaces, respectively, when compared to orally administered TB1 alone. Then, guinea pigs were orally vaccinated with *L. lactis*-TB1-Co1 plus adjuvant CpG-ODN at three different doses, *L. lactis*-TB1-Co1, and PBS. Animals that had been immunized with *L. lactis*-TB1-Co1 plus adjuvant CpG-ODN and *L. lactis*-TB1-Co1 developed elevated antigen-specific serum IgG, IgA, neutralizing antibody, and mucosal SIgA levels, when compared to control groups. Particularly, in mice, *L. lactis*-TB1-Co1 exhibited excellent immune effects than *L. lactis*-TB1. Therefore, *L. lactis*-TB1-Co1 can induce elevations in mucosal as well as systemic immune reactions, and to a certain extent, provide protection against FMDV. In conclusion, M cell-targeting approaches can be employed in the development of effective oral mucosa vaccines for FMDV.

## 1. Introduction

Foot-and-mouth disease (FMD), a clinical acute vesicular and extremely contagious infection, is transmissible among clove hoofed animals [1]. It is caused by the foot-and-mouth disease virus (FMDV), which belong to the *Aphthovirus* genus and *Picornaviridae* family. It is an 8.5 kb positive-sense single-stranded RNA virus [2]. Currently, FMDV O and A serotypes are widespread in China. The FMDV serotype A (A/WH/CHA/09 FMDV) was isolated from the 2009 outbreak in Wuhan, China. Subsequently, from 2009 to 2010, many outbreaks that are closely associated with the above strain have been reported in other regions of China. In 2013, A/GDMM/CHA/2013 FMDV, a new strain, occurred in Maoming, Guangdong, China. FMD is associated with high economic losses [3]. Inactivated vaccines play a fundamental role in controlling FMD outbreaks, especially in developing countries [4]. Although chemically inactivated vaccines provide good protection, they have various limitations, such as their potential for escape of a live virus during production and application and their high production costs [5]. The technology used in multi-epitope proteins as well as their application as vaccines has been widely evaluated [6,7].

Antigenic region structures on capsid surfaces of FMDV have been characterized [8]. An extremely preserved Arg-Gly-Asp (RGD) triplet motif, which is localized on highly mobile exposed G-H loop (140–160 aa) of the capsid protein, VP1, is a key site for viral entry into host cells and is capable of inducing neutralizing antibodies against this virus [9]. On the VP1 of FMDV, the 41–60 aa as well as 200–213 aa are the major neutralizing epitopes that can elicit a strong protective immune response [10,11,12]. Besides, T cell epitopes of FMDV are critical in enhancement of the immunogenicity of peptide vaccines. Collen et al. linearly coupled the T cell epitope (20–40 aa) on VP1 to the G-H loop of VP1 to immunize cattle is a virus challenge experiment [8]. Moreover, T cell epitopes of the 3A protein (21–35 aa) can enhance immune responses to FMDV epitope vaccines [13].

About 90% of pathogenic infections, including FMDV, are transmitted through the mucosal areas. Therefore, mucosal vaccination can help establish a protective immunity against these infections, overcoming the challenges of present injection-based vaccines [14]. Consequently, inoculation of multi-epitope antigens through the mucosal route of entry may improve protective responses against FMDV. Oral mucosal vaccines are advantageous ince their mode of administration is convenient. They are also cost-effective and require simplified logistics with a lower dependence on cold chains during shipments and distribution [15]. However, currently, the commercially accessible oral mucosal vaccines are very few in number. This is attributed to challenges in antigenic delivery into the mucosa and tolerance of the oral mucosal immunity [16]. Thus, to develop efficient oral mucosal vaccines, Microfold cells (M cells) are decent targets for delivering antigens and immune response stimulation [17]. M cells, which are epithelial cells, are localized in Peyer’s patches (PPs) in the intestines, nasopharynx-related lymphoid tissue (NALT), isolated lymphoid follicles, and the appendix. These cells are responsible for the monitoring and transcytosis of antigens, microorganisms, and pathogens [18,19]. M cells play vital roles as gatekeepers of mucosal immunity, even though, in the intestinal tract, only 1 in 10 million epithelial cells is an M cell [20]. Therefore, it is interesting to target antigens to M cells to enhance immune responses. M cell-targeting ligands Co1, obtained through screening of phage display library against in vitro human M-like cells, can promote M cell uptake of oral vaccines and improve antigen-specific immune reactions in systemic as well as mucosal surfaces [21,22].

Currently, *L. lactis* is a good strategy as a delivery vehicle for orally administered mucosal vaccines. *L. lactis,* a model lactic acid bacterium, is generally recognized as safe (GRAS) owing to its longstanding use in human food fermentations and products [23]. This bacterium is used in the development as well as delivery of cytokines and antigens to mucosal surfaces [23]. The nisin-controlled expression (NICE) system has been developed for use in *L. lactis* [24]. This system is the most studied and widely used in expression of exogenous proteins.

We created two recombinant *L. lactis*-TB1 and *L. lactis*-TB1-Co1 as multi-epitope mucosal vaccines, which can express the FMDV multi-epitope antigens, TB1 and TB1-Co1, via the NICE system, respectively. We evaluated their immune effects after oral immunization in mice and guinea pigs.

## 2. Materials and Methods

### 2.1. Animals, Commercial Vaccine, Virus, Bacterial Strain, and Cells

Animals were procured from the experimental animal center at the Lanzhou Veterinary Research Institute (LVRI). The Institutional Animal Care and Use Committee (IACUC) of LVRI approved this study (No. LVRIAEC2018−008). Type A commercial FMDV inactivated vaccine, positive serum from type A FMDV infected swine, and purified inactivated type A FMDV were acquired from OIE/National FMD Reference Laboratory of China. Preserved strains of FMDV, including A/GDMM/CHA/2013, A/WH/CHA/09 and AF72 were acquired from the OIE/National FMD Reference Laboratory of China. *L. lactis* NZ9000 (glycerol bacteria) and *L. lactis* expression plasmid pNZ8148 (dry powder) were bought from MoBiTec, Germany. The *L. lactis* were cultured on an M17 medium (Oxoid, Basingstoke, UK) with 0.5% glucose (GM17) at 30 °C without shaking. Baby Hamster Kidney (BHK) cells were bought from China Center for Type Culture Collection and seeded in the Dulbecco’s modified Eagle’s medium (DMEM, Gibco, Waltham, MA, USA) with 10% fetal bovine serum (FBS, Gibco, Waltham, MA, USA), 100 IU/mL penicillin as well as 100 μg/mL streptomycin in a 5% CO_2_ atmosphere at 37 °C.

### 2.2. Multi-Epitope Gene Design and Synthesis

Signal peptide *SP*usp45 was introduced into the recombinant protein to promote the secretion of the proteins into the culture medium [25]. Promiscuous T helper (Th) cell epitopes were employed in development of immunogenic recombinant multi-epitope proteins [26]. The G-H loop domain as well as VP1 epitopes from the FMDV strains, including A/GDMM/CHA/2013, A/WH/CHA/09 and AF72 (Table 1) were linked by spacer sequences (GGGGS)_2_, as shown in Table 2. Synthesis of the two DNA sequences that encode the multi-epitope proteins (Table 2) was performed by Nanjing GenScript Biotech Co., Ltd. (Nanjing, China) These sequences were codon optimized with *L. lactis,* followed by their insertion into a pUC57 vector with *NcoI* as well as *HindIII* restriction sites on 5′ and 3′ ends, respectively, of multi-epitope genes. Finally, recombinant plasmids pUC57-TB1 and pUC57-TB1-Co1 were obtained. Recombinant plasmids were sequenced for verification. In addition, nucleotide sequences of recombinant proteins and oral adjuvant as shown in Table 3.

### 2.3. Plasmids Were Introduced into L. lactis NZ9000 through Electrotransformation

The TB1 gene was obtained from the pUC57-TB1 plasmid and inserted in corresponding sites of the expression plasmid, pNZ8148, to yield the recombinant expression plasmid, pNZ8148-TB1. Through the same method, we obtained the recombinant expression plasmid, pNZ8148-TB1-Co1. In our previous study, were described the preparation process of NZ9000 competent cells [28]. After melting NZ9000 competent cells on ice, 50 μL of recombinant expression plasmids were added followed by mixing with ice-cold competent cells (100 μL), and placed for 5 min on mice. Transformation of plasmids into competent cells was carried out by 0.2 cm cuvettes in a Gene Pulser electroporator (Bio-Rad, USA) at 200 Ω, 25 µF, and 2.5 kV. The duration of a single electrical pulse was 1.5–5 ms. Cuvettes were immediately mixed with 1 mL of GM17 broth (minus antibiotics but with 20 μL MgCl_2_ and 2 μL CaCl_2_) and placed on ice for 10 min. Incubation was carried out at 30 °C, minus shaking. Then, recombinant *L. lactis* strains were grown and selected on GM17 agar medium with chloramphenicol (10 μg/mL). Finally, we obtained two recombinant strains *L. lactis*-TB1 and *L. lactis*-TB1-Co1.

### 2.4. Expression Levels of Recombinant Proteins

To obtain the TB1 and TB1-Co1 recombinant proteins, strains were grown in GM17 broth with chloramphenicol (10 μg/mL), induced at OD_600_ 0.4 using nisin (5 ng/mL) at 30 °C followed by collection at OD_600_ 1.0. Resuspension of harvested cells was performed in PBS after which preparation of the recombinant proteins was carried out after ultrasonic decompositions for analysis by 12% sodium dodecyl sulphate-polyacrylamide gel electrophoresis (SDS-PAGE). Confirmation of protein specificity was carried out by western blot using the swine anti-FMDV antibodies (1:1000 dilution) and a horseradish-peroxidase (HRP)-conjugated goat anti-swine IgG antibody (1:10,000 dilution; *Abbkine* Scientific Co., Ltd., Wuhan, China). Visualization used ECL-chemiluminescentkit (ECL-plus, Thermo Scientific, Pittsburgh, PA, USA).

### 2.5. Mice Immunization

Bacterial cells (NZ9000/pNZ8148, *L. lactis*-TB1 and *L. lactis*-TB1-Co1) in GM17 broth with chloramphenicol (10 μg/mL), were induced at OD_600_ 0.4 using nisin (5 ng/mL) at 30 °C. Cells were collected at OD_600_ 1.0, washed using PBS and attuned to 1 × 10^9^ colony forming units per milliliter (CFUs/mL) prior to inoculation.

Specific pathogen-free (SPF) female BALB/c mice (125 in number, aged 7 weeks) were randomized in 5 groups of 25 each (Table 4). Each animal in the first four groups was orally immunized by gavage. Immunizations were conducted at three time points (days 1, 11, and 21) for three successive days at every time point. Mice in group E were immunized by intramuscular injection on day one, followed by one booster immunization at a week interval.

Blood samples were obtained by randomly selecting three experimental mice from each group at various time points (that is, on days 10, 20, 30, 37, 44 and 51 after immunization). Then, 300 μL of blood for serum testing was obtained from the angular vein. After mice had been sacrificed by an overdose of ether anesthesia, intestinal mucus samples in 200 μL of sterilized PBS as well as lung lavage liquid specimens in 300 μL of sterilized PBS (three mice per group every time) were obtained using cotton swabs into 1.5 mL centrifuge tubes with 0.01 M EDTA-Na_2_, followed by repeated rinsing.

### 2.6. Immunization of Guinea Pigs and Challenge

Cells (NZ9000/pNZ8148, *L. lactis*-TB1 and *L. lactis*-TB1-Co1) grown at 30 °C in GM17 broth with 10 μg/mL of chloramphenicol, were induced at OD_600_ 0.4 using nisin (5 ng/mL) and harvested at OD_600_ 1.0. They were washed and resuspended in PBS. Before inoculation, final cell concentrations were adjusted to 1 × 10^10^ CFU/mL. Thirty-five FMDV-specific antibody negative female *guinea pigs* (230 to 250 g in weight) were randomized into seven groups of five guinea pigs each (Table 5). Guinea pigs in groups D, E, and F were treated with CpG ODN (5 μg/guinea pig). Immunization procedures were the same as those of mice. Then, for serum tests, 800 μL of blood samples were obtained by cardiac puncture on days 0 (pre-immune day), 10, 20, and finally 30. Saliva and anal swab samples were collected at different time points (0 d, 5 d, 7 d, 10 d, 15 d, 17 d, 20 d, 25 d, 27 d and 30 d) after immunization and kept at −80 °C for successive analyses.

All guinea pigs were challenged with 0.2 mL of 100 GPID_50_ (50% guinea pig infective doses) of FMDV strain AF72 with a subcutaneous injection in the left rear foot at 30 dpv. Guinea pigs in every group were observed for a total of 10 days and clinical symptoms recorded. Assessment of protection of challenged guinea pigs was performed by the absence of FMD-associated lesions, except at the injection site, and vice versa.

### 2.7. Detection of IgG and IgA

Levels of antibodies against FMDV in sera were assessed by ELISA. In brief, coating of 96-well microplates for 12 h was carried out using inactivated whole-virus antigen of FMDV at 4 °C and washed 5 times using PBST. Then, well blocking was performed for at 37 °C for 2 h using 100 μL/well 5% skim milk in PBST. Wells were washed five times using PBST. Serum samples (1:50 dilution) from guinea pigs or mice were added (100 μL/well) and incubated at 37 °C for 1 h. Plates were washed five times using PBST. For mouse serum samples, HRP-conjugated rabbit anti-mouse IgG antibody (1:1000 dilution) or HRP-conjugated sheep anti-mouse IgA antibody (1:10,000 dilution) were added (100 μL/well). For serum samples from guinea pigs, HRP-conjugated rabbit anti-guinea pig IgG (1:1000 dilution) or goat anti-guinea pig IgA (1:10,000 dilution) were added (100 μL/well). Incubation of the microplates was performed for 1 h at 37 °C, washed five times using PBST after which the TMB single-component substrate solution was added (50 μL/well). Incubation of the plates for 15 min was then performed at 37 °C. At last, stop solution (50 μL/well; 2 M H_2_SO_4_) was used to terminate the reactions. Optical absorbance for antibody reactivity was determined at 450 nm within 15 min via a microplate reader.

### 2.8. Detection of SIgA

Detection of SIgA antibody concentrations against FMDV in mice and guinea pig samples was carried out by ELISA. Mice samples included intestinal and lung lavage fluid, while guinea pig samples included saliva and anal swabs. ELISA assays were conducted as previously described.

### 2.9. Viral Neutralizing Antibody Tests (VNT)

Serum specimens from guinea pigs were obtained on day 30 after first immunization and evaluated via viral-neutralizing antibody tests (VNTs) using BHK-21 cells. After 30 min of inactivation of the serum samples at 56 °C, the double ratio dilution method (from 1:2 to 1:256) was used to add 50 μL serum samples to each well of 96-well plates with DMEM (Gibco, Loughborough, UK) with 2% FBS. The serum samples from guinea pigs that had been orally immunized with PBS or *L. lactis* NZ9000/pNZ8148 were used as controls. In a 96--well plate, serum samples were two-fold serial diluted (50 μL/well), after which an equal volume of FMDV AF72 solution (200 TCID_50_/100 μL) was added to the plates at 37 °C for 1 h. Then, addition of BHK cells (100 μL; 10^6^ cells/mL) to the antibody-virus mixture was followed by incubation in a 5% CO_2_ environment at 37 °C for 72 h. Based on the Karber method, neutralizing antibody titers were assessed as reciprocal log_10_ of highest dilutions that could exert protective effects to 50% of the cells against cytopathic effects (CPE).

### 2.10. Analysis of Cytokine Levels

Given that Th cell epitopes contained in recombinant multi-epitope proteins have been shown to upregulate the expression of some cytokines to enhance cellular immune responses, levels of IL-2, IFN-γ, IL-4, IL-10 and IL-5 in mice serum samples were assessed by cytokine ELISA kits (Mlbio, Shanghai, China) as described by the manufacturer.

### 2.11. Determination of T Lymphocyte Subsets

Guinea pig anticoagulated blood samples were obtained on day 30 after initial immunization and stained for 30 min using APC-conjugated rat anti-mouse CD3 antibody, RPE-conjugated rat anti-mouse CD4 antibody and FITC-conjugated rat anti-mouse CD8 antibody (AbD Serotec, Oxford, UK) at room temperature (RT). Evaluation of gated CD3 positive events was carried out for CD3^+^CD8^+^ as well as CD3^+^CD4^+^ T cells. Flow cytometry (BD Biosciences, San Jose, CA, USA) was conducted, followed by analysis by the FlowJo software (FlowJo_V10).

### 2.12. Splenic Lymphocyte Proliferation Assays

On the 30th day after first immunization of mice, isolation of splenic lymphocytes was performed using a Spleen Lymphocyte Separation Medium (Solarbio, Beijing, China). Assessment of cell proliferation levels was carried out by the MTT Lymphocyte Proliferation Assay Kit (Solarbio, Beijing, China). In brief, suspension of splenic lymphocytes was carried out in RPMI-1640 medium supplemented with FBS (10%; Gibco, Waltham, MA, USA) and 1% penicillin/streptomycin (Gibco, Waltham, MA, USA). Then, they were culivated in 96-well plates (5 × 10^6^ cells per mL and 50 μL/well). Sample stimulation was carried out using 50 μL inactivated FMDV antigens (10 μg/mL; specific antigen stimulation), concanavalin A (ConA, 10 μg/mL) (positive control) and culture medium (negative control), respectively, incubated for 72 h in 5% CO_2_ at 37 °C. Proliferation responses were detected by an MTT Lymphocyte Proliferation Assay Kit (Solarbio, Beijing, China). Findings are expressed as stimulation indices (SI, ratio of stimulated/unstimulated samples at OD_490_ nm).

### 2.13. Statistical Analysis

Data are shown as mean ± SD. GraphPad Prism 7 (LaJolla, CA, USA) was used for statistical analyses. Two-way ANOVA, *t* tests and multiple *t* tests were performed to determine significant differences among and between means, respectively. *p* ≤ 0.05) was significant and *p* ≤ 0.01 was very significant.

## 3. Results

### 3.1. Expression of Multi-Epitope Proteins by L. lactis

Table 2 shows the scheme for construction of FMDV multi-epitope recombinant proteins, TB1 as well as TB1-Co1, which were the immunogens are shown in Table 2. Recombinant proteins were expressed as soluble proteins in *L. lactis* NZ9000 using the NICE system. Strangely, we found that there was no expression of the recombinant proteins in the culture medium samples. However, we found the expression of the proteins of interest in the lysed cell supernatant. In this case, we think it may be due to the long induction time for the bacteria in the culture flask at 30 °C, which leads to proteins degradation. Schematic diagram of recombinant proteins TB1 and TB1-Co1 as shown in Figure 1A. Molecular mass for each of the two proteins was about 40 kDa (Figure 1B). Detection of recombinant proteins was carried out using anti-FMDV (serotype A) antibodies and Western blot (Figure 1C).

### 3.2. Antibody Responses

To investigate recombinant multi-epitope protein-induced mucosal immune responses in mice, ELISA was performed to measure FMDV-specific IgA levels in serum samples and FMDV-specific SIgA levels in intestinal as well as lung fluid samples obtained on days 10, 20, 30, 37, 44, and 51 after the priming immunization. FMDV-specific IgA levels during the study period are shown in Figure 2. IgA antibody levels in groups C and D continuously increased and peaked on day 30, but gradually decreased afterwards. Moreover, serum IgA antibody levels in group D were significantly elevated when compared to group C (*p* < 0.05). Differences in serum IgA levels for groups A and B were not significant on day 10, 20 and 51(Figure 2A). Meanwhile, elevations in serum IgA antibody levels in group E were not significant. Comparable findings for FMDV-specific SIgA levels were obtained for intestinal fluid (Figure 2B) and lung lavage fluid samples (Figure 2C) in immunized mice. However, SIgA antibody levels in intestinal fluid from group D were elevated when compared to those in lung lavage fluid of group D and the case for group C was the same. Systemic immunity in mice was assessed by measuring FMDV-specific IgG serum levels (Figure 2D). Groups C, D, and E exhibited anti-FMDV serotype A IgG antibodies. After initial and boost immunizations in all groups, over time elevations in serum IgG levels were significant, except for groups A and B. In addition, special anti-FMDV IgG antibody levels in group E were significantly elevated when compared to the other four groups (*p* < 0.001). Importantly, group D serum IgG levels were significantly elevated when compared to group C, except day 10. (*p* < 0.05). Therefore, orally administrered *L. lactis*-TB1-Co1 can elicit stronger mucosal and systemic immune responses when compared to *L. lactis*-TB1. Furthermore, recombinant *L. lactis*-induced mucosal immune responses were strong, when compared to inactivated vaccine-induced responses, since the inactivated vaccine hardly induced an effective mucosal immune response.

In guinea pigs, mucosal immune responses were assessed by quantifying FMDV-specific IgA levels in serum and FMDV-specific SIgA levels in anal swab and saliva samples. As shown in Figure 3, there were no significant differences among groups on day zero (*p* > 0.05). During the growth period, serum IgA antibody levels for groups C, D, E, and F showed marked elevations from day 10 to 30. However, serum IgA antibody levels for the remaining groups did not exhibit significant changes. Furthermore, differences in serum IgA levels for the three groups (C, D and F) were not significant, but IgA levels in the three groups were significantly elevated when compared to group E (*p* < 0.001) (Figure 3A). Comparable findings for FMDV-specific SIgA levels were acquired from anal swabs (Figure 3B) and saliva samples (Figure 3C) in immunized guinea pigs. However, SIgA antibody levels were elevated in group D, relative to group C (*p* < 0.05). In addition, SIgA antibody levels in anal swab samples were significantly elevated on day five while SIgA antibody levels in saliva samples on day seven were elevated. Moreover, SIgA antibody levels in anal swab and saliva samples in groups D and F were higher than that of group E (*p* < 0.001), and there were significant differences in groups D and F (*p* < 0.05), which indicates *L. lactis*-TB1-Co1 induced the production of SIgA antibodies in a dose-dependent manner in guinea pigs. The FMDV-associated systemic immune reactions in guinea pigs were evaluated by determining serum IgG levels (Figure 3D). On day zero, IgG levels among groups were not significant (*p* > 0.05). IgG antibody levels were elevated and peaked on day 30. Although FMDV-specific IgG levels in group D were elevated, relative to group C, differences were not significant (*p* > 0.05). Groups D and F exhibited significantly elevated FMDV-specific IgG levels when compared to group E during the whole period (*p* < 0.001). Therefore, *L. lactis*-TB1-Co1 could induce guinea pigs to produce mucosal and systemic immune responses. Moreover, the oral adjuvant, CpG ODN, enhanced mucosal immune responses, but could not enhance systemic immune responses. Mucosal responses developed early when compared to systemic responses.

### 3.3. T Lymphocyte Levels

The peripheral blood T lymphocytes in mice 30 days after the first immunization are shown in Figure 4. The abundance of CD3^+^, CD3^+^CD4^+^, and CD3^+^CD8^+^ T lymphocytes is a vital indicator of specific immunity. Assessment of peripheral blood CD3^+^, CD3^+^ CD4^+^ and CD3^+^CD8^+^ T lymphocyte changes was performed by Flow cytometry (Figure 4A). After oral immunization with *L. lactis*-TB1-Co1, as shown in Figure 4B, the percentage of CD3^+^ T lymphocytes significantly increased than other groups (*p* < 0.05). Besides, the group C was higher than groups A and B in CD3^+^ T lymphocytes (*p* < 0.05). Moreover, oral immunization with *L. lactis*-TB1-Co1 induced a higher percentage of CD3^+^CD4^+^ T lymphocytes than the groups B and C (*p* < 0.05). However, there was no significant difference (*p* > 0.05) between the groups D and E (Figure 4C). Furthermore, the trend of CD3^+^CD8^+^ T lymphocytes was similar with that of CD3^+^CD4^+^ (Figure 4D). However, the percentage of CD3^+^CD8^+^ T lymphocytes in group D significantly increased than group E (*p* < 0.05). These findings imply that *L. lactis*-TB1-Co1 induced stronger cellular and humoral immune responses than *L. lactis*-TB1, with humoral immune responses being predominant.

### 3.4. T Cell Proliferative Responses in Spleen

Specific lymphocyte proliferations in spleen were evaluated by the MTT colorimetric assay. As shown in Figure 5, for mice in groups C and D, lymphocytes exhibited significantly high proliferation levels compared to groups A and B (*p* < 0.05; *p* < 0.001). Group D exhibited significantly high lymphocyte proliferation levels following treatment with inactivated FMDV antigen, relative to group C (*p* < 0.05). These results suggest that *L. lactis*-TB1-Co1 promoted cellular immune responses against FMDV when compared to *L. lactis*-TB1.

### 3.5. Cytokine Concentrations in Recombinant L. lactis Administered Mice

We determined serum IL-4, IL-5, IL-10, IFN-γ, and IL-2 concentrations in recombinant *L. lactis* administered mice on days 10, 20, 30, 37, 44, and 51. As shown in Figure 6, all the other cytokine levels continuously increased and reached their peaks at day 30 and subsequently decreased, except for the case of IL-10 which peaked on day 37 (Figure 6A–E). Concentrations of IL-4, IL-5, and IL-10 in group D mice were significantly elevated compared to group C at day 30 (*p* < 0.001). Compared to mice in group C, IFN-γ titers in group D were significantly elevated on days 30 (*p* < 0.05). In addition, IL-2 levels in groups D were markedly elevated (*p* < 0.01), relative to those in group C at day 30. Notably, IL-5, IFN-γ and IL-2 levels in group D were markedly elevated than those in group C on day 51(*p* < 0.001). Furthermore, IL-4, IL-10, IFN-γ as well as IL-2 levels in group B were significantly elevated on day 30 when compared to those of group A (*p* < 0.001). Therefore, *L. lactis*-TB1-Co1 can induce effective humoral and cellular immune responses. Moreover, live carrier *L. lactis* NZ9000/pNZ8148 may play an accessory function to some extent in humoral as well as cellular immune responses.

### 3.6. Neutralizing Viral Antibodies

Neutralizing antibody levels correlate with protection from FMDV, therefore, serum anti-viral neutralizing antibody titers from guinea pigs were measured. Figure 7 shows that serum neutralizing antibody titers from vaccine group animals were markedly elevated than groups A and B animals. No significant differences of the neutralizing antibody titers were observed between the group C and D (*p* > 0.05). This result implies that the oral adjuvant, CpG ODN, may not enhance the production of neutralizing antibodies in guinea pigs. Moreover, the neutralizing antibody titers in groups D and F were higher than that of group E (*p* < 0.05; *p* < 0.01), but there were no significant differences in groups D and F (*p* > 0.05), which indicates *L. lactis*-TB1-Co1 induced the production of antibodies without a dose-dependent manner in guinea pigs. However, neutralizing antibody levels in group G were significantly elevated when compared to other groups. Therefore, *L. lactis*-TB1-Co1 stimulated a potent humoral immune response without a dose-dependent manner while CpG ODN is an oral adjuvant in guinea pigs, even though its efficacy is low compared to the inactivated vaccine.

### 3.7. Protective Effects against FMDV Challenge

Protective efficacies of recombinant multi-epitope proteins were assessed in immunized guinea pigs on day 30 after the first immunization and their clinical symptoms (swollen soles and secondary vesicles) documented daily for 10 days. Table 6 shows that all of the five guinea pigs in groups A, B, and E developed lesions on both of their rear feet following viral challenge, while there were no clinical symptoms of FMD for animals in group G. Interestingly, although guinea pigs in groups A, B, and E were not protected, clinical symptoms in group E were less severe than those in the other two groups. In addition, *L. lactis*-TB1-Co1 alone or with the oral adjuvant, CpG ODN, used to immunize guinea pigs in groups C, D, and F exhibited identical protection rates against FMDV challenge (60%), indicating that the oral adjuvant, CpG ODN, could not improve protection in guinea pigs. Therefore, the multi-epitope mucosal vaccine, *L. lactis*-TB1-Co1, is a probable vaccine candidate, with a protection rate of 60% against 100 GPID_50_ virulent viral challenge in guinea pigs.

## 4. Discussion

Traditional inactivated vaccines induce strong humoral immune responses and weak cell-mediated immune responses [29]. Moreover, it could not prevent viral entry through the mucosa [30]. Generally, FMDV infections occur at mucosal surfaces, therefore, mucosal immunization using specific multi-epitope antigens can protect against infections. In addition, mucosal IgA was associated with protective effects among FMDV-infected pigs, indicating the necessity for establishment of oral mucosa vaccines against FMDV [31]. Therefore, oral mucosa vaccines against FMDV using multi-epitope antigens may overcome the limitations associated with injection vaccines, and can significantly activate the first line immune defenses in the gut. Thus, intestinal mucosa immune systems, which forms the first barrier against bacterial, viral and parasitic intestinal infections is being evaluated [17]. SIgA is very important on mucosal surfaces, therefore, SIgA secretion is vital for efficient antigen-specific mucosal immune responses [32]. As a first line immune component, SIgA protects via intracellular neutralization, immune exclusion, and antigen excretion. Immune exclusion is mediated via antigenic or bacterial entrapment and suppression of their interactions with epithelial cells [33].

In mucosal lymphoid tissues, vaccine antigens are essential for generation of antigen-specific SIgA [34]. In the mucosal epithelium, macromolecule transcytosis occur in M cells via specific or non-specific receptor-associated processes [35]. This, studies have aimed at evaluating immune strategies that target antigens to M cells to develop efficient oral mucosa vaccines. The interaction between M cell-targeting ligand Co1 and C5aR on M cells is associated with antigen delivery as well as initiation of antigen-specific immune responses [21]. In oral mucosal immunization, since the antigen is degraded in the gastric juice, the antigen can be presented with a capsule or a protective carrier. *L. lactis* was used as a live carrier to present antigenic proteins. Moreover, *L. lactis* exerts pro-inflammatory effects and responses in the intestinal mucosa, making it a good oral vaccine [36]. Induction of TB1 and TB1-Co1 protein expressions in *L. lactis* was carried out using the NICE system. Multi-epitope antigen proteins were identified in bacterial lysates and not culture supernatants, implying intracellular expressions in *L. lactis*. We found that the Co1-conjugated multi-epitope antigen, TB1 of FMDV, had effective interactions with M cells of ileal PPs relative to TB1 alone minus Co1 ligand conjugation. In addition, in this study, the oral mucosal vaccine, *L. lactis*-TB1-Co1, against FMDV was developed and experimented in mice and guinea pigs. We found that *L. lactis*-TB1-Co1 triggered systemic, mucosal, as well as cell-mediated immune reactions.

Mucosal immune reactions, especially IgA, are efficient against mucosal surface pathogens [37]. *L. lactis*-TB1-Co1 administration in mice led to elevations in SIgA titers in intestinal and lung lavage fluids. However, intestinal SIgA levels were elevated, compared to lungs. In addition, IgG antibodies revealed that *L. lactis*-TB1-Co1 triggered high-level systemic immune responses. Therefore, mucosal immunization can initiate mucosal as well as circulating antibody responses. This outcome can be attributed to various reasons. One, there was an abundance of M cells and dendritic cells in the follicle epitheliums of gut-related lymphoid tissues [38]. These cells transported antigenic proteins, TB1-Co1, from gut lumens to systemic immune cells, which stimulated systemic immune responses. Two, common mucosal immune system (CMIS) has an integrated cross-communication pathway of lymphoid tissues that are made up of inductive as well as effector sites for host protection against pathogens. Major CMIS effector molecules are IgA, IgG antibodies as well as cytokines. These effector molecules can home to system-related lymphoreticular tissues via blood circulation [39]. Comparable findings were obtained in guinea pigs. Moreover, we found that IgG antibody secretions occurred later than SIgA, which may play an important role against pathogen invasion from mucosal surfaces. However, the involved mechanisms are uncertain. As previously reported [40], intramuscular inoculation with inactivated vaccines, regardless of the assay, does not induce IgA responses, consistent with our findings. This highlights the need to understand immune responses to this virus, so as to develop new, rapid-action vaccines for FMDV.

Various pathogens are capable of inducing cytokine secretion by CD4^+^ T cells. Cytokines, which are small-molecule proteins with many biological activities, are produced by both immune as well as some non-immune cells [41]. As cell signaling molecules, they regulate immune responses, participate in immune cell differentiation and development, mediate inflammatory responses, stimulate hematopoiesis, and participate in tissue repairs. Auxiliary signals for B cell proliferation as well as differentiation are obtained from CD4^+^ T cells, while CD8^+^ T cells are cytotoxic killer cells and innate immune regulators [42]. Effector CD4^+^ T cells include Th1 as well as Th2-specific responses. IL-2 and IFN-γ, which are produced by Th1 cells, enhance the activation as well as proliferation of CTL, macrophages, and NK cells, promote pathogen cytotoxicity and phagocytosis, and enhance CD8^+^ T cell and innate immune responses [43]. IL-10 and IL-4, which are produced by Th2 cells enhance B cell proliferation as well as differentiation to plasma cells, thereby initiating antibody secretion and promoting humoral immune responses. Th1 responses, such as IFN-γ and IL-2 secretion are very effective against intracellular pathogens. Other Th2 responses, including IL-5, IL-4, and IL-10 secretions can efficiently eliminate pathogens from body fluids. In this study, *L. lactis*-TB1-Co1 enhanced Th1- as well as Th2-type responses, consistent with other studies [44,45]. This is due to the fact that Th cell epitopes are vital in the promotion of both humoral as well as cell immune responses by stimulating T cells [26,46]. T cell-secreted cytokines can also enhance the proliferation as well as differentiation of B cells into plasma cells [47]. Cellular immunity effectively protects against FMD [48]. Cellular immune responses are assessed by lymphocyte proliferation *in vitro*. Con A stimulates T cell proliferation. Our results demonstrated that *L. lactis*-TB1-Co1 regulated specific T lymphocyte proliferative responses, and secreted cytokines to enhance the abundance of CD4^+^ and CD8^+^ T lymphocytes in mice.

Challenge test results showed that guinea pigs immunized with *L. lactis*-TB1-Co1 exhibited 60% protection rates. These findings demonstrate that, to some extent, oral immunization with *L. lactis*-TB1-Co1 protected against FMDV. To promote the effects of immunization of recombinant *L. lactis*, we used an efficient oral mucosal adjuvant, CpG-ODN [49,50]. However, it was found that guinea pigs orally vaccinated with *L. lactis*-TB1-Co1 plus CpG-ODN did not exhibit significant protection rates. In this regard, we speculate that there are two reasons for the situation. On the one hand, it is possible that the dose of CPG ODN used is low in this experiment, which makes it unable to effectively activate B cells, monocytes, macrophages and dendritic cells to secrete antibodies, various cytokines and chemokines [50]. On the other hand, it may not be effective in guinea pigs due to differences in species. For higher protection effects, double doses of *L. lactis*-TB1-Co1 were administered, we found a consistent result. Interestingly, half of the dose used did not protect the guinea pigs. We found that protection rates did not increase when the vaccination volume was doubled. This outcome may be due to the ability of M cells to present antigens had reached its limit. However, it inhibited disease severity and delayed the onset of clinical symptoms, relative to negative-control animals. Thus, *L. lactis*-TB1-Co1 may have led to the production of specific neutralizing antibody titers. As previously reported [26,51,52], there was not a strong association between neutralizing antibodies and protection. It has been proven that interferon-mediated cellular immunity can protect pigs from FMDV infection [53,54]. Thus, to evaluate the efficacy of vaccines for FMD, in addition to in vitro neutralizing activities, cellular immune responses, antibody affinities and other antibody responses, should be considered [26].

As an exploratory research, this study has several limitations. First, since no purified multi-epitope antigen proteins were obtained, in vitro and in vivo experiments on antigen uptake were not performed. Second, there is no currently unified gold standard for evaluating mucosal immunity. Therefore, we adopted traditional challenge methods, which may weaken protection efficiencies of this oral mucosal vaccine. We postulate that better experimental results may be obtained if the challenge is carried out by oral or aerosol methods. In future studies, we will adopt safe and cheap mucosal adjuvants to achieve better experimental results. More studies are needed to assess the cross-protection that is mediated by oral mucosal vaccines against other FMDV serotype A lineages in guinea pigs and swine, which will better help us understand the effectiveness of the vaccine.

In conclusion, we have demonstrated that M cell-targeting ligand Co1 is an important adjuvant for FMDV oral mucosal vaccines. It is important in efficient delivery of ligand-conjugated multi-epitope antigens to mucosal immune components and effective initiation of mucosal and systemic immune responses in mice. Moreover, we showed that CpG-ODN, as the oral mucosal vaccine adjuvant, could enhance mucosal immune responses and systemic immune responses of guinea pigs. Therefore, although oral vaccination with *L. lactis*-TB1-Co1 could, to some extent, protect against FMDV challenge, it suppressed disease severity and delayed clinical symptoms, implying that *L. lactis*-TB1-Co1 has potential as a novel oral mucosal vaccine for FMD.

## Figures and Tables

**Figure 1 viruses-13-02036-f001:**
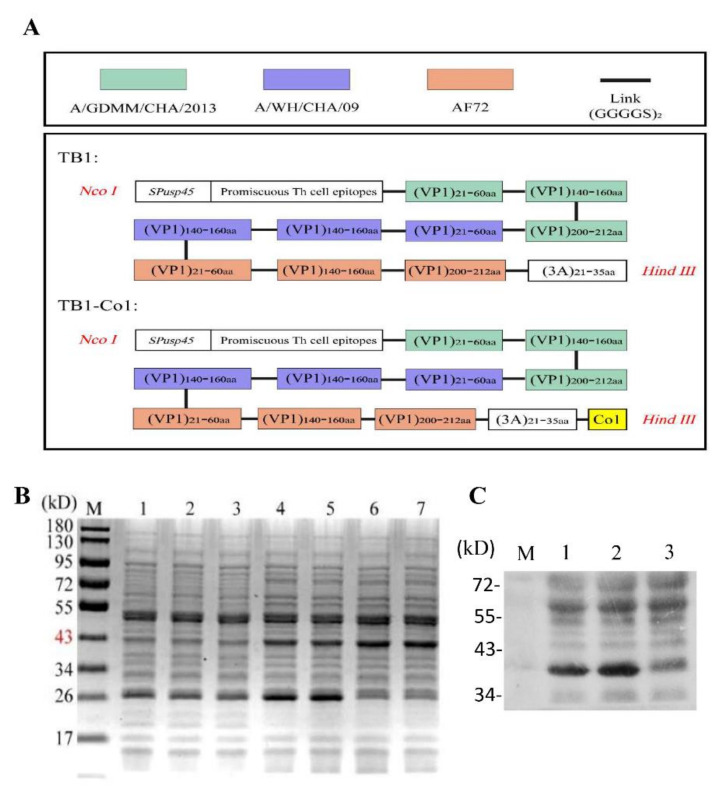
Design and evaluation of recombinant proteins in *L. lactis* NZ9000. (**A**) Schematic diagram of recombinant proteins design. (**B**) SDS--PAGE analysis of the two multi-epitope proteins. M: Protein marker; Lane 1: NZ9000/pNZ8148 with nisin as the control; Lane 2: *L. lactis*-TB1 without nisin; Lane 3: *L. lactis*-TB1-Co1 without nisin; Lane 4–5: *L. lactis*-TB1 with nisin; Lane 6–7: *L. lactis*-TB1-Co1 with nisin. (**C**) Western blot analysis of the fusion proteins. Lane 1: *L. lactis*-TB1 with nisin; Lane 2: *L. lactis*-TB1-Co1 with nisin; Lane 3: NZ9000/pNZ8148 with nisin.

**Figure 2 viruses-13-02036-f002:**
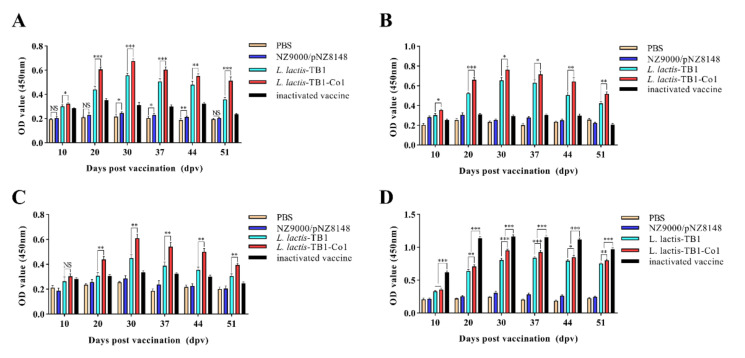
Levels of FMDV-specific antibodies in mice. (**A**) IgA levels in serum on days 10, 20, 30, 37, 44 and 51. (**B**) SIgA antibody levels in intestinal fluid on days 10, 20, 30, 37, 44 and 51. (**C**) SIgA antibody levels in lung lavage fluid on days 10, 20, 30, 37, 44 and 51. (**D**) IgG antibody levels in serum on days 10, 20, 30, 37, 44 and 51. Data are shown as mean ± SD. Analyses were carried out by multiple *t* tests (NS = *p* > 0.05; * *p* < 0.05; ** *p* < 0.01; *** *p* < 0.001). Ig, immunoglobulin; SD, standard deviation.

**Figure 3 viruses-13-02036-f003:**
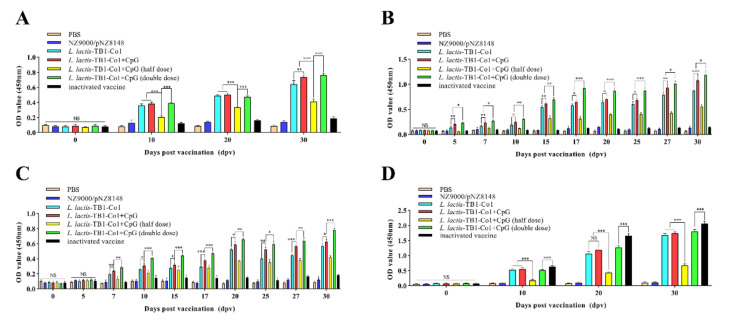
Titers of FMDV-specific antibodies in guinea pigs. (**A**) IgA levels in serum on days 0, 10, 20, and 30. (**B**) SIgA levels in anal swabs on days 0, 5, 7, 10, 15, 17, 20, 25, 27, and 30. (**C**) SIgA levels in saliva on days 0, 5, 7, 10, 15, 17, 20, 25, 27, and 30. (**D**) IgG antibody levels in serum on days 0, 10, 20 and 30. Data are shown as mean ± SD. Analyses were performed by multiple *t* tests (NS = *p* > 0.05; * *p* < 0.05; ** *p* < 0.01; *** *p* < 0.001).

**Figure 4 viruses-13-02036-f004:**
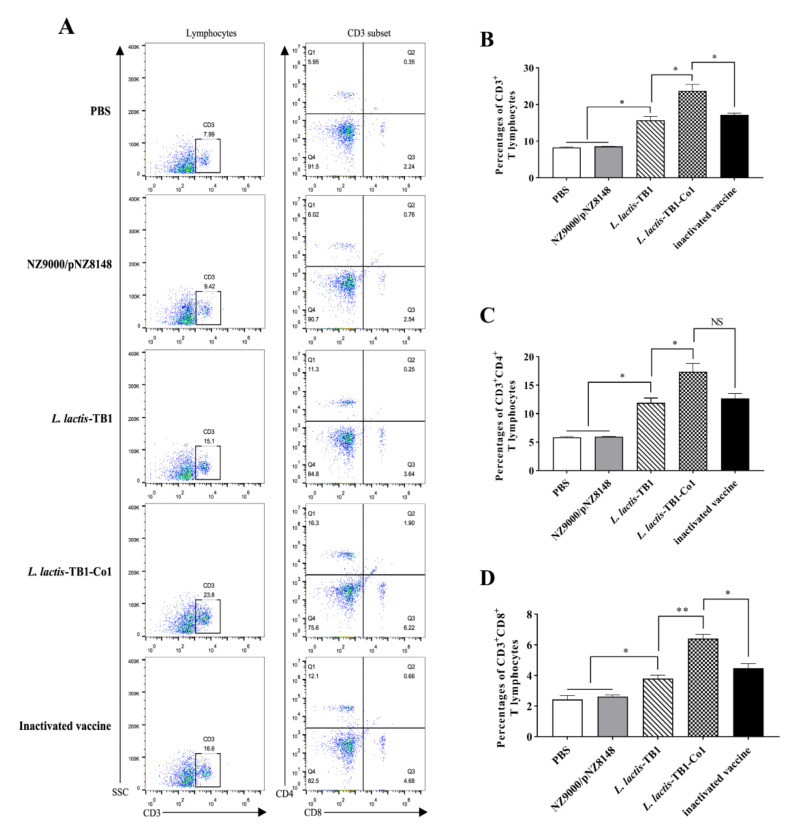
Percentage of CD3^+^, CD3^+^CD4^+^, and CD3^+^CD8^+^ T lymphocytes in mice peripheral blood at day 30. (**A**) The representative flow cytometry plots showing CD3^+^, CD3^+^CD4^+^ and CD3^+^CD8^+^ T lymphocytes in the peripheral blood. (**B**) Frequencies of CD3^+^ T lymphocytes. (**C**) Frequencies of CD3^+^CD4^+^ T lymphocytes. (**D**) Frequencies of CD3^+^CD8^+^ T lymphocytes. Data are presented as mean ± SD. Analyses were performed by *t* tests (NS = *p* > 0.05; * *p* < 0.05; ** *p* < 0.01).

**Figure 5 viruses-13-02036-f005:**
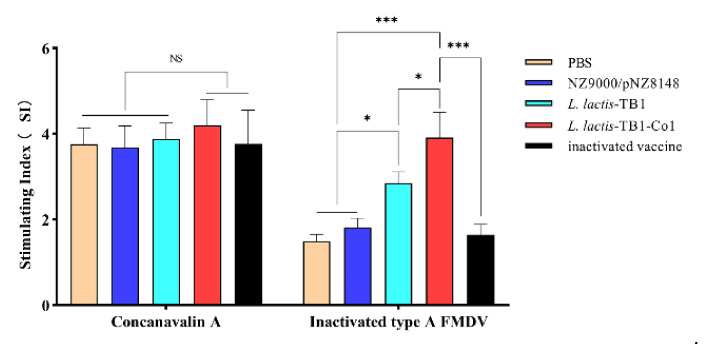
T lymphocyte proliferation levels. Splenic lymphocytes were obtained on day 30 post-vaccination and respectively induced with inactivated serotype A FMDV and concanavalin (ConA). Lymphocyte proliferation levels were evaluated by MTT colorimetric analysis. Stimulation index (SI) shows the ratio of OD_490_ nm of stimulated wells to OD_490_ nm of unstimulated wells. Data are presented as mean ± SD. Analyses were performed by two-way ANOVA test (NS = *p*>0.05; * *p* < 0.05; *** *p* < 0.001).

**Figure 6 viruses-13-02036-f006:**
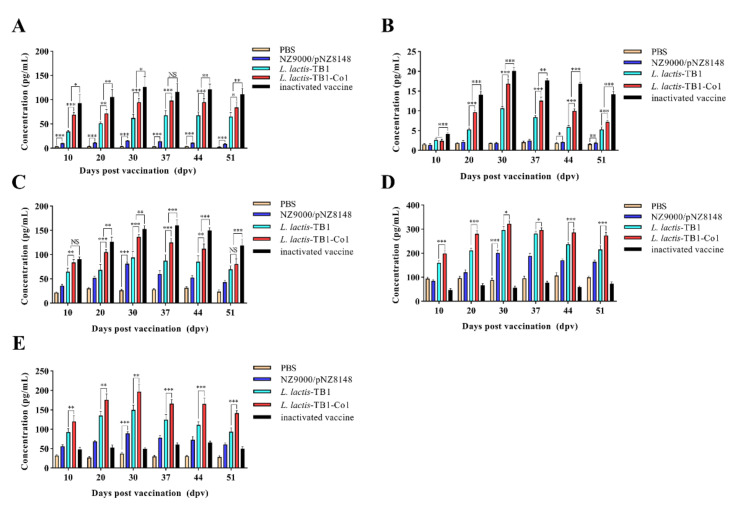
Detection of immune responses in mice serum samples obtained on days 10, 20, 30, 37, 44 and 51 after oral vaccination. (**A**) IL-4. (**B**) IL-5. (**C**) IL-10. (**D**) IFN-γ. (**E**) IL-2 were determined by indirect ELISA. Data are expressed as means ± SD. Analyses were performed by multiple *t* tests (NS = *p* > 0.05; * *p* < 0.05; ** *p* < 0.01; *** *p* < 0.001).

**Figure 7 viruses-13-02036-f007:**
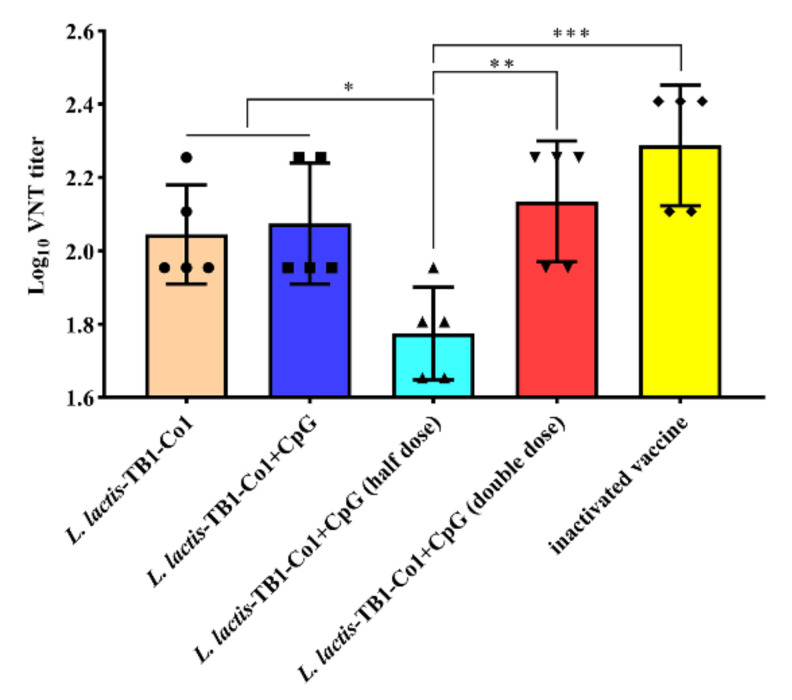
Titers of FMDV-specific neutralizing antibodies in guinea pigs. Data were calculated as log_10_ of max serum dilution that neutralized 100 TCID_50_ of FMDV strain AF72. Error bars show SD. Analyses were performed by *t* tests (NS = *p* > 0.05; * *p* < 0.05; ** *p* < 0.01; *** *p* < 0.001).

**Table 1 viruses-13-02036-t001:** G-H loop of the VP1 protein from FMDV serotype A strains and other short peptides.

Name	Sequences	Origin of Sequences
*SP* _Usp45_	MKKKIISAILMSTVILSAAAPLSGVYA	pVE5523 [25]
Promiscuous Th cell epitopes	ISISEIKGVIVHKIETILF	[26]
VP1_21–60_	ETQVQRRYHTDVGFLMDRFVQIKPVGPTHVIDLMQTHQHG	A/GDMM/CHA/2013
VP1_140–160_	QNRRGDLGPLAARLAAQLPAS	G-H loop of A/GDMM/CHA/2013
VP1_200–212_	HKQKIIAPAKQLL	A/GDMM/CHA/2013
VP1_21–60_	ETQVQRRHHTDVSFIMDRFVQIKPVSPTHVIDLMQTHQHG	A/WH/CHA/09
VP1_140–160_	ATRRGDLGSLAARLAAQLPAS	G-H loop of A/WH/CHA/09
VP1_21–60_	ETQVQRRQHTNVGFIMDRFVKIPSQSPTHVIDLMQTHQHG	AF72
VP1_140–160_	NAGRRGDLGSLAARVAAQLPA	G-H loop of AF72
VP1_200–212_	RHKQRIIAPAKQL	AF72
3A_21–35_	AAIEFFEGMVHDSIK	[27]
Co1	SFHQLPARSPLP	M cell-targeting ligand [21]

**Table 2 viruses-13-02036-t002:** Components of recombinant proteins.

Epitopes.	Amino Acid Sequences
TB1	*SP*_Usp45_-Promiscuous Th cell epitopes-(GGGGS)_2_-A/GDMM/CHA/2013 VP1_21–60_-(GGGGS)_2_-A/GDMM/CHA/2013 VP1_140–160_-(GGGGS)_2_-A/GDMM/CHA/2013 VP1_200–212_-(GGGGS)_2_-A/WH/CHA/09 VP1_21–60_-(GGGGS)_2_-A/WH/CHA/09 VP1_140–160_-(GGGGS)_2_-A/WH/CHA/09 VP1_140–160_-(GGGGS)_2_-AF72 VP1_21–60_-(GGGGS)_2_-AF72 VP1_140–160_-(GGGGS)_2_-AF72 VP1_200–212_-(GGGGS)_2_-3A_21–35_
TB1-Co1	*SP*_Usp45_-Promiscuous Th cell epitopes-(GGGGS)_2_-A/GDMM/CHA/2013 VP1_21–60_-(GGGGS)_2_-A/GDMM/CHA/2013 VP1_140–160_-(GGGGS)_2_-A/GDMM/CHA/2013 VP1_200–212_-(GGGGS)_2_-A/WH/CHA/09 VP1_21–60_-(GGGGS)_2_-A/WH/CHA/09 VP1_140–160_-(GGGGS)_2_-A/WH/CHA/09 VP1_140–160_-(GGGGS)_2_-AF72 VP1_21–60_-(GGGGS)_2_-AF72 VP1_140–160_-(GGGGS)_2_-AF72 VP1_200–212_-(GGGGS)_2_-3A_21–35_-(GGGGS)_2_-Co1

**Table 3 viruses-13-02036-t003:** Nucleotide sequences of recombinant proteins and oral adjuvant.

Name	Nucleotide Sequences
TB1	5′ATGAAAAAGAAAATTATTTCAGCAATTCTTATGTCTACAGTTATTTTAAGTGCTGCAGCTCCACTTTCAGGTGTTTATGCTATTAGTATTTCAGAAATTGGTAAAGTTATTGTTAAACATATTGAAGGAATTTTTCTTTTGGGAGGAGGTGGATCAGGTGGAGGTGGATCTGAAACACAAGTTCAACGTAGATATCATACTGATGTTGGATTTTTAATGGATCGTTTTGTTCAAATTAAACCAGTTGGACCTACACATGTTATTGATCTTATGCAAACTCATCAACATGGTGGAGGTGGAGGTAGTGGAGGTGGAGGTTCACAAAATCGTAGAGGAGATTTAGGTCCATTAGCAGCTAGATTAGCAGCTCAATTACCTGCATCAGGTGGAGGTGGAAGTGGTGGTGGAGGTTCACATAAACAAAAAATTATTGCACCTGCTAAACAACTTTTGGGTGGTGGAGGTTCTGGAGGTGGAGGTAGTGAAACTCAAGTACAACGTAGACATCATACTGATGTTTCTTTTATTATGGATAGATTTGTACAAATTAAACCAGTTAGTCCAACTCATGTAATTGATTTAATGCAAACACATCAACACGGTGGAGGAGGAGGTTCTGGAGGAGGTGGATCTGCAACAAGACGTGGAGATTTAGGTTCATTAGCAGCTCGTTTAGCAGCTCAATTACCAGCAAGTGGAGGTGGAGGATCTGGTGGAGGTGGATCTGCTACAAGAAGAGGAGATTTAGGTTCTTTAGCTGCACGATTGGCAGCTCAATTACCTGCTAGTGGAGGTGGAGGAAGTGGAGGTGGTGGTTCAGAAACTCAAGTACAAAGAAGACAACATACTAATGTTGGTTTTATTATGGATCGTTTTGTTAAAATTCCATCTCAAAGTCCAACTCATGTTATTGATTTAATGCAGACACATCAACACGGAGGTGGAGGTGGTTCAGGTGGTGGAGGTTCAAATGCAGGACGTCGTGGTGATTTAGGAAGTTTAGCAGCTAGAGTTGCAGCTCAATTACCAGCTGGTGGAGGTGGTTCAGGAGGTGGAGGTAGTCGTCATAAACAACGTATTATTGCTCCTGCTAAACAATTGGGAGGTGGAGGATCTGGAGGAGGTGGTTCTGCAGCTATTGAATTTTTCGAAGGAATGGTTCATGATAGTATTAAA-3′
TB1-Co1	5′ATGAAAAAGAAAATTATTTCAGCAATTCTTATGTCTACAGTTATTTTAAGTGCTGCAGCTCCACTTTCAGGTGTTTATGCTATTAGTATTTCAGAAATTGGTAAAGTTATTGTTAAACATATTGAAGGAATTTTTCTTTTGGGAGGAGGTGGATCAGGTGGAGGTGGATCTGAAACACAAGTTCAACGTAGATATCATACTGATGTTGGATTTTTAATGGATCGTTTTGTTCAAATTAAACCAGTTGGACCTACACATGTTATTGATCTTATGCAAACTCATCAACATGGTGGAGGTGGAGGTAGTGGAGGTGGAGGTTCACAAAATCGTAGAGGAGATTTAGGTCCATTAGCAGCTAGATTAGCAGCTCAATTACCTGCATCAGGTGGAGGTGGAAGTGGTGGTGGAGGTTCACATAAACAAAAAATTATTGCACCTGCTAAACAACTTTTGGGTGGTGGAGGTTCTGGAGGTGGAGGTAGTGAAACTCAAGTACAACGTAGACATCATACTGATGTTTCTTTTATTATGGATAGATTTGTACAAATTAAACCAGTTAGTCCAACTCATGTAATTGATTTAATGCAAACACATCAACACGGTGGAGGAGGAGGTTCTGGAGGAGGTGGATCTGCAACAAGACGTGGAGATTTAGGTTCATTAGCAGCTCGTTTAGCAGCTCAATTACCAGCAAGTGGAGGTGGAGGATCTGGTGGAGGTGGATCTGCTACAAGAAGAGGAGATTTAGGTTCTTTAGCTGCACGATTGGCAGCTCAATTACCTGCTAGTGGAGGTGGAGGAAGTGGAGGTGGTGGTTCAGAAACTCAAGTACAAAGAAGACAACATACTAATGTTGGTTTTATTATGGATCGTTTTGTTAAAATTCCATCTCAAAGTCCAACTCATGTTATTGATTTAATGCAGACACATCAACACGGAGGTGGAGGTGGTTCAGGTGGTGGAGGTTCAAATGCAGGACGTCGTGGTGATTTAGGAAGTTTAGCAGCTAGAGTTGCAGCTCAATTACCAGCTGGTGGAGGTGGTTCAGGAGGTGGAGGTAGTCGTCATAAACAACGTATTATTGCTCCTGCTAAACAATTGGGAGGTGGAGGATCTGGAGGAGGTGGTTCTGCAGCTATTGAATTTTTCGAAGGAATGGTTCATGATAGTATTAAAGGAGGTGGAGGATCAGGTGGAGGTGGTTCATCTTTTCATCAATTACCAGCTAGATCTCCATTACCT-3′
CpG ODN	5′-TCGCGACGTTCGCCCGACGTTCGGTA-3′

**Table 4 viruses-13-02036-t004:** Immunization schedule in mice model.

Groups	Number	Immunization Dose	Ingredient	Immunization Mode
A	25	0.2 mL	PBS	Oral
B	25	0.2 mL (10^9^ CFU/mL)	NZ9000/pNZ8148	Oral
C	25	0.2 mL (10^9^ CFU/mL)	*L. lactis*-TB1	Oral
D	25	0.2 mL (10^9^ CFU/mL)	*L. lactis*-TB1-Co1	Oral
E	25	0.2 mL	inactivated vaccine	Intramuscular

CFU, colony forming unit.

**Table 5 viruses-13-02036-t005:** Immunization schedule in *guinea pig* models.

Groups	Number	Immunization Dose	Ingredient	Immunization Mode
A	5	0.2 mL	PBS	Oral
B	5	0.2 mL (10^10^ CFU/mL)	NZ9000/pNZ8148	Oral
C	5	0.2 mL (10^10^ CFU/mL)	*L. lactis*-TB1-Co1	Oral
D	5	0.2 mL (10^10^ CFU/mL)	*L. lactis*-TB1-Co1+CpG	Oral
E	5	0.1 mL (10^10^ CFU/mL)	*L. lactis*-TB1-Co1+CpG	Oral
F	5	0.4 mL (10^10^ CFU/mL)	*L. lactis*-TB1-Co1+CpG	Oral
G	5	0.4 mL	Inactivated vaccine	Intramuscular

CFU, colony forming unit.

**Table 6 viruses-13-02036-t006:** Protection rates in guinea pigs after 100 GPID_50_ FMDV serotype A challenge.

Group	Primary Vesicles	Secondary Vesicles	Protection Rate (%)
A	5/5	5/5	0
B	5/5	5/5	0
C	5/5	2/5	60
D	5/5	2/5	60
E	5/5	5/5	0
F	5/5	2/5	60
G	5/5	0/5	100

## Data Availability

The datasets used and/or analyzed in this study are obtained and available from the corresponding authors upon a reasonable request.

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
