# Peer review of "Immune Responses to Orally Administered Recombinant Lactococcus lactis Expressing Multi-Epitope Proteins Targeting M Cells of Foot-and-Mouth Disease Virus"

_viruses, 2021, doi:10.3390/v13102036_

Round 1

Reviewer 1 Report

This reviewer commends for their interesting study.

 I suggest that the authors use different colors for data visualization for all the illustrations/figures/graphs from pages 8-12.  There appear to be effective vaccines  against FMD Virus. In fact,  there is a new patent-pending process  which claims to be a breakthrough for Foot-and-Mouth Disease Vaccine1. The authors need to convince policy makers, practitioners, and the scientific community how their novel vaccine could potentially be more effective than existing vaccines and/or those pending the patent process.

Reference

  1. https://www.newswise.com/articles/new-patent-pending-process-is-a-breakthrough-for-foot-and-mouth-disease-vaccine

Author Response

Dear Ms. Nomi Ouyang and reviewer,

Thank you for your letter and for the reviewer's insightful comments concerning our manuscript entitled "viruses-1390431". According with your advice, we amended the relevant part in the manuscript. Some of your comments were answered below.

Point 1: I suggest that the authors use different colors for data visualization for all the illustrations/figures/graphs from pages 8-12. 

Response 1: Thank you for your suggestion. We have revised all the figures in the revised manuscript according to your comments. Thank you for your valuable comments, making our works more like works of art. We have uploaded revised manuscript as a Word file. Please see the attachment.

Point 2: The authors need to convince policy makers, practitioners, and the scientific community how their novel vaccine could potentially be more effective than existing vaccines and/or those pending the patent process.

Response 2: Thank you very much for the reference that you have provided, so that I can learn more about it. By reading the reference carefully, I found that the novel vaccines in the literature have many advantages compared to inactivated vaccines. However, compared with the oral vaccine in our study, our vaccine has some advantages that it cannot achieve. For example, a. Because it is an oral vaccine, it can be applied on a large scale to feed-intensive farms through drinking water or feed; b. Since it is taken orally, avoid the stress response to animals that can be injected with needles; c. Vaccination is very convenient, neither requires training of specialized talents nor syringes, thus saving a lot of manpower, material resources and financial resources. In summary, we believe that these advantages are very important for convincing policy makers, practitioners, and the scientific community to adopt our new vaccines.

In all, once again, thank you very much for your comments and suggestions.

Reviewer 2 Report

This study explores the Immune effects of oral immunization with genetically modified Lactococcus lactis expressing multi-epitope proteins targeting M cells of foot-and-mouth disease virus in mice and guinea pigs. The work is quite interesting, clear, and well-organized. The introduction provides enough background. The aim of the study is clear. The methods are described in detail. The Results and Discussion are correctly described. I only have some minor observations that are listed in the following lines.

  1. Line 99: add relevant protocol number.
  2. Line 149: specify.
  3. Line 182: Rephrase.
  4. Figure 1B is not clear enough. Also, add labels to the bands.
  5. Figure 4 is hard to read.
  6. Add a schematic diagram to summarize the experimental design.
  7. The provided supplementary figures are not labeled and hence, not useful.
  8. Add a list of abbreviations.

Author Response

Dear Ms. Nomi Ouyang and reviewer,

Thank you for your letter and for the reviewer's insightful comments concerning our manuscript entitled "viruses-1390431". Now we have revised this paper exactly according to your comments, and found these comments are very helpful. We hope this revision can make our article more acceptable. The revisions were addressed point by point below.

Point 1: Line 99: add relevant protocol number. 

Response 1: Line 99,“(No.LVRIAEC2018−008)” was added. 

Point 2: Line 149: specify.

Response 2: Line 149, the statements of ” Visualization was done using a chemiluminescent substrate reagent” was corrected as “Visualization used ECL-chemiluminescentkit (ECL-plus,Thermo Scientific)”.

Point 3: Line 182: Rephrase.

Response 3: Line 182-183, the statements of “Left rear legs of guinea pigs were subcutaneously administered with 0.2 mL of 100 GPID50 (50% guinea pig infectious dose) of AF72 at 30 days after initial immunization” was revised as “All guinea pigs were challenged with 0.2mL of 100 GPID50 (50% guinea pig infective doses) of FMDV strain AF72 with a subcutaneous injection in the left rear foot at 30 dpv”. 

Point 4: Figure 1B is not clear enough. Also, add labels to the bands.

Response 4: Thank you very much for this suggestion. For this, I would like to explain why this happened. Because the primary antibody is a self-made porcine poly-antiserum and the bacterial proteins in this experiment have not been purified, there are many mixed bands that is blurred in the western blot. In addition, the WB bands in this study are obtained by exposing them with a film pressed in a dark room. After exposing, we go out of the dark room and then mark the size of the protein marker in the film. Finally, we have submitted the original figures of WB and replaced the figure with more clear and labeled in the submission system and hope you can check them out.

Point 5: Figure 4 is hard to read.

Response 5: It is really true as you suggested that. This is my negligence to submit a screenshot of Figure 4 in the manuscript. Based on your suggestion, I replaced it with the original image with higher pixels. 

Point 6: Add a schematic diagram to summarize the experimental design.

Response 6: The schematic diagram to summarize the experimental design was added to Figure 1. 

Point 7: The provided supplementary figures are not labeled and hence, not useful.

Response 7: We submitted supplementary figures that were high pixel and labeled again. In addition, we have replaced all the pictures in the revised manuscript.  

Point 8: Add a list of abbreviations.

Response 8: Line 11, “Abbreviations: FMDV, foot and mouth disease virus; NICE, nisin-controlled expression; L. lactis, Lactococcus lactis; IgG, immunoglobulin G; SIgA, secretory immunoglobulin A; IL-2, interleukin-2; IFN-γ, interferon gamma; IL-4, interleukin-4; IL-5, interleukin-5; IL-10, interleukin-10; M cell, microfold cell; NALT, nasopharynx-related lymphoid tissue; GRAS, generally recognized as safe; PPs, peyer's patches” was added.

In all, we think your comments are very helpful for improving our manuscript. Thank you and review again for your help!

Reviewer 3 Report

Thank you for the opportunity to review this manuscript. This manuscript reported the development of engineering Lactococcus lactis expressing multi-epitope antigens derived from Foot and mouth disease virus fused with M cells targeting peptide for oral mucosal vaccine systems against FMDV infections. This manuscript is well written, and the information here might be helpful for readers. However, the authors need to address some concerns before publication. Please see my specific comment stated below.

Major Comments

  1. Line 87: Please explain briefly about GRAS for readability.
  2. Table 2: Please add nucleotide sequences.
  3. The authors have fused the Co1 peptide to the C-terminal side of TB1, but are they also considering adding it to the N-terminal side? If you are considering it, please add the data. If not, please describe the rational reason for adding the Co1 peptide sequence to the C-terminal side.
  4. Is the dosage of lactis optimized? Please describe the rationale for this dosage.
  5. Is the dosage of CpG ODN optimized? Please describe the rationale for this dosage.
  6. Please provide details such as sequence information of CpG ODNs used in this study.
  7. It is desirable to add data for TB1 plus CpG as well as TB1-Co1 plus CpG.
  8. Antigen-specific antibody levels should be indicated by endpoint titer, not absorbance.
  9. The authors have designed the plasmid so that the antigen is expressed as a secreted protein, but please make it clear in the text.
  10. 2B: Western blot image data should be presented as a whole membrane including molecular weight markers, not as a cropped image. Also, please include the nisin-free group in the data.
  11. Figs 2, 3, 4, 6, and 7: Please describe the statistical methods in the legends.
  12. 5: A t test is not appropriate. Please re-assess using ANOVA test.
  13. Discussion: Please discuss the reason why the addition of CpG ODN in the vaccine formulation did not alter the vaccine effects.
  14. The major concern of this manuscript is that the authors did not discuss the rational reason why does the oral mucosal vaccine system used in this study have a weak protective effect against infection even though it predominantly enhances the production of antigen-specific mucosal sIgA compared to inactivated vaccine which i.m. received?

Minor comments:

  1. Line 64: There is an unnecessary space inserted between “can” and “enhance”. Please delete it.

Author Response

Dear Ms. Nomi Ouyang and reviewer,

Thank you for your letter and for the reviewer's comments concerning our manuscript entitled "viruses-1390431". Those comments are all valuable and very helpful for revising and improve our paper, as well as the important guiding significance to our researches. We have studied those comments carefully and have made correction which we hope meet with approval.

Revised portion are marked in red in the paper. The main corrections in the paper and responds to the reviewer's comments are as following:

Point 1: Line 87: Please explain briefly about GRAS for readability.

Response 1: L. lactis, a model lactic acid bacteria, is Generally Recognized As Safe (GRAS). Lactococcus lactis, which have a generally regarded as safe (GRAS) status owing to their longstanding use in human food fermentations and products1

Line 88, “owing to their longstanding use in human food fermentations and products” was added.

Reference1: DOI:10.1146/annurev-food-022510-133640

Point 2: Table 2: Please add nucleotide sequences.

Response 2: The nucleotide sequences of TB1 and TB1-Co1 were added to Table 3 in revised manuscript. 

Point 3: The authors have fused the Co1 peptide to the C-terminal side of TB1, but are they also considering adding it to the N-terminal side? If you are considering it, please add the data. If not, please describe the rational reason for adding the Co1 peptide sequence to the C-terminal side.

Response 3: This is an excellent and insightful question. We considered adding Co1 peptide to the N-terminal side of TB1 at the beginning, but we did not design this multi-epitope sequences in the end. When looking up the reference1, I found a very interesting study. Their research found that Co1 conjugation to the C-terminus of EDIII enhanced EDIII-specific immune responses in systemic and mucosal compartments by T-cell stimulation and also found oral immunization with EDIII-Co1 maximally induced not only neutralizing antibody against all DENV serotype but also a Th2/Th17-skewed immune response. Especially, SPLs prepared after systemic challenge of the antigen from EDIII-Co1A-immunized mice proliferated efficiently against in vitro antigen stimulation. However, when Co1 is conjugated to the N-terminus of EDIII, they observed inability of Co1-EDIII in M-cell targeting was unexpected. Therefore, they assumed that the inability of Co1-EDIII-mediated in vivo M-cell targeting was closely related with protein instability, since it is known that anchoring of a protein in its C-terminus generally increases protein stability through reducing flexibility of the protein2. In view of the above reasons, we did not design the multi-epitope sequences added Co1 peptide to the N-terminal side of TB1 in this study.

Reference 1: DOI:10.1093/intimm/dxt029

Reference 2: DOI:10.1371/journal.pone.0016226 

Point 4: Is the dosage of lactis optimized? Please describe the rationale for this dosage.

Response 4: In this study, we did not optimize the dosage of L. lactis,since Liu et al1. have done similar experiments in our research group.

Reference 1: https://doi.org/10.1007/s10529-020-02900-6  

Point 5: Is the dosage of CpG ODN optimized? Please describe the rationale for this dosage.

Response 5: In this study, we did not optimize the dosage of CpG ODN. By reading the reference1, we know that 1ug of CpG ODN as an oral adjuvant can have a good effect in mice. Through consideration, we finally determined that the amount of CpG ODN is 5ug per guinea pig as an exploratory research.

Reference 1: DOI:10.1016/s0264-410x(00)00215-2 

Point 6: Please provide details such as sequence information of CpG ODNs used in this study.

Response 6: The nucleotide sequences (5’-TCGCGACGTTCGCCCGACGTTCGGTA-3’) of CpG ODN were added to Table 3 in revised manuscript.  

Point 7: It is desirable to add data for TB1 plus CpG as well as TB1-Co1 plus CpG.

Response 7: Thanks for your suggestion. The data for TB1-Co1 plus CpG have been presented to readers in the guinea pig experiment, but the data for TB1 plus CpG have not. We think this way, because it has been proved that TB1-Co1 is better than TB1 in the mice, we only studied TB1-Co1 in the guinea pigs. In addition, the main purpose of the study is to develop a more effective novel oral vaccine, so we did not do the data for TB1 plus CpG in the guinea pigs. Our next plan is to verify the effect of TB1-Co1 plus CpG on pigs in a biosafety level 3 laboratory. If conditions permit, we will add TB1 plus CpG in the study.

Point 8: Antigen-specific antibody levels should be indicated by endpoint titer, not absorbance.

Response 8: Thanks for your suggestion. In this study, antigen-specific antibody levels be indicated by absorbance, which was mainly based on the research paper of Song et al.

Reference: https://pubmed.ncbi.nlm.nih.gov/31227355/.

Point 9: The authors have designed the plasmid so that the antigen is expressed as a secreted protein, but please make it clear in the text.

Response 9: Thank you so much for your suggestion. Actually, we did construct a secreted recombinant plasmids in this study. As we all know, signal peptide SPusp45 was introduced into the recombinant proteins to promote the secretion of the proteins into the culture medium. However, after the inducer was added to the bacterial solution in the logarithmic growth phase to induce expression for four hours, the bacteria and the culture medium were collected. We found that there was no expression of the recombinant proteins in the culture medium samples, that is, we did not detect the protein of interest is secreted into the medium. Fortunately, we found the expression of the proteins in the lysed cell supernatant. In this case, we think it may be due to the long induction time for the bacteria in the culture flask at 30 degrees Celsius, which leads to proteins degradation. Therefore, the experimental animals were inoculated with the induced bacteria in this study.

Line 112, “Signal peptide SPusp45 was introduced into the recombinant protein to promote the secretion of the proteins into the culture medium.” was added.

Line 258, “Strangely, we found that there was no expression of the recombinant proteins in the culture medium samples. However, we found the expression of the proteins of interest in the lysed cell supernatant. In this case, we think it may be due to the long induction time for the bacteria in the culture flask at 30°C, which leads to proteins degradation.” was added.

Point 10: 2B: Western blot image data should be presented as a whole membrane including molecular weight markers, not as a cropped image. Also, please include the nisin-free group in the data.

Response 10: This is a very good suggestion. We have replaced this figure in revised manuscript. As for why there is no nisin-free group in the figure, our explanation is as follows.1) Because adding the inducer nisin to the bacteria culture will slow down the growth of the bacteria, and the recombinant proteins cannot be purified in this study, it is difficult to guarantee the quantification when loading the sample.2) In the animal experiments of this study, all groups were added with the same amount of inducer nisin (5ng/mL) except the PBS group and the inactivated vaccine group. 3) In our research group, the previous research results of Liu et al1 showed that the results of NZ9000/pNZ8148 are almost the same regardless of whether the inducer nisin is present or not. In view of the above reasons, we used NZ9000/pNZ8148 with inducer nisin as the control group in this study.

Reference1: https://doi.org/10.1007/s10529-020-02900-6

Point 11: Figs 2, 3, 4, 6, and 7: Please describe the statistical methods in the legends.

Response 11: We described the statistical methods in the legends of the figures you mentioned. In this study, the data in Figure 2, 3, and 6 were analyzed through t test, but Figure 4 and 7 used t tests analysis. The above modifications have been added to the legends of the corresponding figures in revised manuscript. 

Point 12: 5: A t test is not appropriate. Please re-assess using ANOVA test.

Response 12: Thank you for your suggestion. We re-analyzed the data in Figure 5 using the Two-way ANOVA test, and made corresponding replacements in the revised manuscript. 

Point 13: Discussion: Please discuss the reason why the addition of CpG ODN in the vaccine formulation did not alter the vaccine effects.

Response 13: This is an excellent question. We are very sorry for our negligence of this point. In this regard, we speculate that there are two reasons for the situation. On the one hand, it is possible that the dose of CPG ODN used is low in this experiment, which makes it unable to effectively activate B cells, monocytes, macrophages and dendritic cells to secrete antibodies, various cytokines and chemokines. On the other hand, it may not be effective in guinea pigs due to differences in species. According to your suggestion, we have added this discussion part to the revised manuscript at line 500. 

Point 14: The major concern of this manuscript is that the authors did not discuss the rational reason why does the oral mucosal vaccine system used in this study have a weak protective effect against infection even though it predominantly enhances the production of antigen-specific mucosal sIgA compared to inactivated vaccine which i.m. received?

Response 14: This is also really an excellent question, and thank you for reading my article carefully. Regarding the question you mentioned, we made a certain explanation at line 515-518. As you know, SIgA produced by an animal's body is the first defensive barrier against the entry of external pathogens into the body, which is great significance to the health of the animal. However, in this study, although relatively high SIgA was produced, the live viruses were directly inoculated with a subcutaneous injection in the left rear foot of guinea pigs during the challenge. This method has actually greatly reduced the protective effect of SIgA on the body. Therefore, we got a lower protection rate. This really makes it very difficult for us to truly evaluate the effectiveness of mucosal vaccines using traditional methods of challenge. We need an innovative method that everyone recognizes. Unfortunately, this method has not been invented so far.

Point 15: Line 64: There is an unnecessary space inserted between “can” and “enhance”. Please delete it.

Response 15: Line 64, “the unnecessary space” was deleted.

We tried our best to improve the manuscript and made some changes in the manuscript. These changes will not influence the content and framework of the paper. And here we did not list the changes but marked in red in revised paper.

We appreciate for Editor and Reviewer' warm work earnestly, and hope that the correction will meet with approval. Once again, thank you very much for your comments and suggestions.

Round 2

Reviewer 3 Report

I have gone through the revised manuscript, and I think all my comments have been adequately addressed. Thank you again for the opportunity to review this manuscript.